

# Effectiveness of a modified Balint group process on empathy and psychological skills employing Kirkpatrick's evaluation framework

Jumana Antoun[1,*], Beatrice Khater[1,*], Hala Itani[1], Jihane Naous[1,2] and Maya Romani[1]

[1] Family Medicine, American University of Beirut, Beirut, Lebanon
[2] Community Health and Family Medicine, University of Florida, Florida, United States
* These authors contributed equally to this work.

## ABSTRACT

**Background:** To evaluate, using Kirkpatrick's evaluation model, a modified Balint group (BG) by adding 5–10 min at the end of the session, where the facilitators will debrief the residents about critical themes mentioned in the session.

**Methods:** A quasi-experimental study with a mixed-method design was conducted among family medicine residents over 1 year, using focus groups and validated tools filled by residents and their corresponding patients. The residents' empathy through self and patient evaluation, psychological skills, and satisfaction with the educational activity were measured.

**Results:** The focus group revealed that the residents were aware of the change and considered it a closure to the encounter, helping decrease some uncertainty. Most of the residents did not consider the change helpful. Using validated instruments, BG was ineffective at improving the residents' empathy and psychological skills. There was a statistically significant main effect of time on Psychological Medicine Inventory (PMI) scores, $F_{(1,13)} = 7.709$, $p = 0.016$.

**Conclusion:** Debriefing at the end of BG by the facilitators about key themes may help give the residents closure, decrease the uncertainty, and make them more aware of their feelings. Nevertheless, Balint groups are still not very well accepted by the residents, and they prefer direct feedback and support groups.

# INTRODUCTION

The Balint group (BG) was named after Michael Balint, a British psychoanalyst, who established safe and regular sessions for family physicians to reflect on a medical case that touched them and preoccupied their minds outside the clinic encounter (*Balint, 1955*). A BG often includes a small group of physicians or residents in training and is facilitated by a trained leader. The goal of the BG is to provide a better understanding of the emotional content of the doctor-patient relationship by physicians to make them more capable physicians (*Yazdankhahfard, Haghani & Omid, 2019*). BG enhances the levels of residents'

Corresponding author
Maya Romani, mr39@aub.edu.lb

self-reported psychological medicine skills (*Turner & Malm, 2004*), contributes to better satisfaction and a higher sense of control at work (*Kjeldmand, Holmstrom & Rosenqvist, 2004*), and encourages residents' development toward a professional identity as a physician (*Torppa et al., 2008*). BG was found to increase empathy scores among medical students using self-rated Jefferson's School Empathy Scale (*du Vaure et al., 2017*). However, some controversy exists on whether BG increases the empathetic abilities of the residents or not (*Airagnes et al., 2014*; *Ghetti, Chang & Gosman, 2009*), in addition to its effect on reducing the risk for burnout (*Bar-Sela, Lulav-Grinwald & Mitnik, 2012*; *Ghetti, Chang & Gosman, 2009*) and emotional intelligence (*Antoun et al., 2020*).

Furthermore, there is an ongoing discussion on the best methodology to study the effects of Balint. Although most of these studies were quantitative (*Torppa et al., 2008*), some argue that qualitative studies may provide more significant information on BG (*Van Roy, Vanheule & Inslegers, 2015*). Another common methodological challenge is an assessment of the Balint outcomes from the patient's perspective. For example, patient perspective may be as crucial as self-assessment in measuring empathy among physicians (*Bernardo et al., 2019*), yet, the majority of the studies measure the physicians' point of view only (*Bernardo et al., 2018*).

Balint's original description of the leader's role included no teaching or guidance in the sessions (*Merenstein & Chillag, 1999*). There was also an emphasis on the goal of the BG seminars to learn to live with uncertainty and focus on self-discovery (*Merenstein & Chillag, 1999*). Many residents expressed strong dissatisfaction as they were uncertain at the end of the sessions (*Antoun et al., 2014*; *Smith & Anandarajah, 2007*). They felt discouraged from solving the problem and had no solutions or correct answers (*Antoun et al., 2014*). Residents expressed their need to be told how to do things at the end of the traditional BG session, but they were not (*Kjeldmand & Holmstrom, 2010*). This had been one of the main barriers to fully benefiting from the conventional BG seminars (*Van Roy, Vanheule & Inslegers, 2015*), sometimes leading to dropouts from the sessions (*Kjeldmand & Holmstrom, 2010*). A reassessment of the design of the traditional Balint seminars is vital.

Many observed Balint sessions did not follow the traditional way described by Balint and were more willing to give the "right answer" and provide teaching and guidance (*Merenstein & Chillag, 1999*). Furthermore, one residency program modified the sessions based on the residents' frustration by (1) encouraging the residents to choose the case discussion or topic and (2) allowing problem-solving (*Smith & Anandarajah, 2007*). Despite the literature about modified Balint, none of the recent studies examined the effectiveness of the modifications (*Merenstein & Chillag, 1999*; *Smith & Anandarajah, 2007*; *Torppa et al., 2008*). This study aims to evaluate a modified BG by adding 5–10 min at the end of the BG, where the facilitators debrief the residents about critical or significant themes mentioned in the session. Kirkpatrick's evaluation model will be used to evaluate the effectiveness of the modified BG regarding residents' satisfaction, empathetic abilities, and psychological skills, using physicians' and patients' perspectives.

## MATERIALS AND METHODS

A quasi-experimental study with a mixed-method design was conducted to measure the effectiveness of a modified BG among family medicine residents over 1 year. BG was modified by adding 5–10 min at the end of the session, where the facilitators debriefed the residents about key themes mentioned in the session. The modified BG's effectiveness was evaluated by measuring the residents' empathy through self and patient evaluation, residents' psychological skills, and satisfaction with the educational activity. The American University of Beirut granted ethical approval (#SB-201900379) for the conduct of the study. All patients provided written informed consent.

The Kirkpatrick Model has been widely used in the evaluation of health profession education. It has been utilized in at least 603 studies, with a steady increase in its use in the last decade (*Allen, Hay & Palermo, 2022*). Kirkpatrick identified four broad classes of program outcomes: (1) the "reaction," measuring the participants' feelings, (2) the "learning," the principles, facts, and techniques that were understood and absorbed by the participants, (3) the "behavior" assessing any change in the practice of the participants, and finally (4) the "results" (*Kirkpatrick, 1996*). First, the "reaction" will be interpreted as the residents' satisfaction-measured qualitatively *via* a focus group. Second, the "learning" was measured in terms of empathetic abilities and psychological skills *via* the Jefferson Scale of Physician Empathy (JSPE) and the Psychological Medicine Inventory (PMI), respectively. Third, the "behavior" was directly measured from the patient's perspective, using the Consultation and Relational Empathy scale (CARE).

### Residents
#### Setting

BG training was introduced at the American University of Beirut in 2011.
All post-graduate years 2, 3, and 4 attend the sessions. Throughout the 2020–2021 academic year, conventional BGs (10 sessions) were delivered in the first 6 months, followed by the modified BG (eight sessions) for the last 6 months.

#### Procedure

A convenience sample included all 18 family medicine residents receiving Balint training. The residents were invited to approach their patients through an email sent by the department residency program coordinator. The research assistant (RA) approached those who accepted, explained the study, and ensured that the residents had signed the informed consent.

The residents were invited to complete a hard copy questionnaire every 2 months and return it to the family medicine department administrator's office. The questionnaire includes demographics (age, gender, marital status, current residency level), the Psychological Medicine Inventory (PMI), which is a set of eleven questions used to assess psychological sensitivity & psychological abilities on a seven-point scale (1 = not at all to 7 = to a great extent) (*Ireton & Sherman, 1988*), and Jefferson Scale of Physician Empathy questionnaire (JSPE), which is a 20 Likert-type item answered on a seven-point scale (1 = strongly disagree to agree 7 = strongly).

The residents joined a 1-h focus group in December 2020 at the end of the traditional BG and in May 2021 at the end of the modified BG. Following an interview guide (Appendix A), the focus groups were led by BK, a faculty member who is not involved in the BG seminars. The focus group discussions were audiotaped and transcribed *verbatim*.

## Patients

The triage nurse approached the patients of residents who agreed to participate in the study. If they were interested, the RA contacted the patient, explained the study, and obtained consent. At the end of the clinical encounter with the resident, the RA handed the patient the questionnaire to fill in a private place in the waiting or triage area. The RA was available for any clarification. The anonymous questionnaire included demographics, type of illness, code of the resident filled by the RA, and the Consultation and Relational Empathy scale (CARE) either in English or Arabic, according to the patient's preference. The Consultation and Relational Empathy scale (CARE) is a patient-completed questionnaire that measures empathy in the context of the therapeutic relationship during a one-on-one consultation between a clinician and a patient. It comprises ten items answered on a five-point scale ranging from 1 = poor to 5 = excellent. The total score is calculated by summing the score of the 10 items.

## Data analysis

For the quantitative part, the data were analyzed with IBM SPSS Statistics (Version 27; SPSS, Inc., Chicago, IL, USA), and the significance level, $\alpha$, was set at 0.05. A two-way repeated-measures ANOVA was used to answer the research question of whether there is a difference in the Jefferson Empathy Scale or PMI based on the training method across time. The skewness and kurtosis values of all the empathy scales at different time points were within ±1.0. The dependent variables were operationalized with the validated Jefferson Scale of Physician Empathy/PMI and an interval variable. Four variables were used: T-beginning-traditional, T-end-traditional, T-beginning-modified, and T-end-modified. The two factors were time (two levels) and training method (two levels).

Null hypothesis 1: There is no difference in the empathy and psychological skills scores between the two training methods.

Alternative hypothesis 1: There is a difference in the empathy and psychological skills scores between the two training methods.

Null hypothesis 2: No difference in the empathy and psychological skills scores between the beginning and end of the year.

Alternative hypothesis 2: There is a difference in the empathy and psychological skills scores between the beginning and end of the year.

Null hypothesis 3: There is no interaction between training method and time affecting the scores of empathy and psychological skills.

Alternative hypothesis 3: There is an interaction between training method and time affecting the scores of empathy and psychological skills.

Two-way ANOVA was used to answer the research question of whether an interaction between the resident and training method affects the CARE score.

Null hypothesis 1: There is no difference in the CARE score between the two training methods.

Alternative hypothesis 1: There is a difference in the CARE score between the two training methods.

Null hypothesis 2: There is no difference in the CARE score among the residents.

Alternative hypothesis 2: There is a difference in the CARE score among the residents.

Null hypothesis 3: There is no interaction between the training method and residents in affecting the CARE score.

Alternative hypothesis 3: There is an interaction between the training method and residents in affecting the CARE score.

Using G\*Power 3.1.9.7 software, a sample size of 77 patients in both the traditional and modified BG periods is needed. The calculation was based on a comparison of mean analysis with an effect size measured by Cohen's f2 of 0.454 with an alpha level of 0.05 and power of 80%. The effect size was calculated based on previous literature showing that the overall mean CARE Measure score was 47.8 with a standard deviation of 4.4 (*Bikker et al., 2017*) and a hypothesized difference of two points set by the researchers as the patients usually give high scores.

For the qualitative data, an inductive approach was used. Both BK and JA individually familiarized themselves with the transcripts and identified codes. The codes were organized to identify a content pattern, which in turn was organized into themes and compressed into a display that facilitates conclusion drawing. JA is one of the Balint leaders. To address reflexivity, JA was always attentive to her bias and vigilant about being nonjudgmental. BK is trained in qualitative methodology and narrative medicine. She is not involved in Balint groups, which has added to the reliability of the coding and avoided personal biases in interpreting the data.

# RESULTS

## Satisfaction

A total of nine residents were present in each focus group. The satisfaction of the residents was measured qualitatively *via* the focus group. The following themes were extracted from the initial focus group post the traditional BGs:

1-Feelings towards Balint were mixed. Negative emotions included: dissatisfaction, lack of enjoyment, frustration, draining, anger, and waste of time, while positive feelings included a sense of confidence, comfort, and relief.

2-The residents did not like the lack of closure of the encounter and left with more questions than started. They considered Balint sessions just questions and feelings with ideas and discussions. Balint seminars are more about questions than solutions.
One resident suggested that facilitators of the Balint seminars give hints at the end of the session.

*"I like the idea of the Balint and the idea of having the space to share those experiences; however, I agree with my colleagues that the current way is frustrating. I feel that I don't have closure."*

*"I want to stay something the fact that we can talk about issues that are going on is good, but maybe we are not dealing with it properly. It is just questions and feelings."*

3-They considered the aim of Balint seminars not clear and theoretical, as they could not see its application. They did not consider Balint seminars helpful and could not notice a change in their behavior with patients. When probed to answer what they thought the aim of Balint seminars, they included: improving patient experience, learning from mistakes and experience, dealing with difficult patients, identifying one's emotions, and controlling emotions.

*"I cannot take it back to the clinic and improve how I deal with the patient. The benefit stops at being a vent-out session and having a space to talk to my colleagues without judging how I feel. But it stops there."*

*"I can see how similar to counseling; when you go to the counselor does not tell you what the answer is, but you are supposed to discover it by yourself, but I feel that part I am not getting what I want in terms of the questions being asked; how should I reflect; asked to put myself in the shoes of the patient; I am not getting that"*

4-They benefited from BG by listening to different opinions, especially from colleagues, instead of attending. They considered BG a nonjudgmental space (ventilation session) to share experiences. It was a space that brought them together and made them a "stronger cohort."

*"Comfort and relief that my colleagues are also experiencing difficult situations and that they are also being put under the same circumstances; this is kind of relief; it is not just me; everyone is facing the same problems; this is a good thing."*

*"It makes us closer to each other and makes us stronger as a cohort. This is really important."*

5-Few reported that it benefited their relationship with future patients and gave them a new perspective. Few acknowledged that the benefit might be long-term and subconscious.

*"make you think about it but needs long years of experience and thinking and redirecting your way; it is a long way, not something to be achieved in one or two sessions."*

*"Sometimes it helps with long term; when we discuss a case, the questions asked make us think about how we can deal with the patient better in the future."*

*"I can not say this encounter went well because I got to benefit from Balint; maybe it is something subtle, but I am not feeling this"*

*"Throughout the residency training, I have changed with my patient's encounters and how I deal with them; I guess I can not say that Balint per se and the session influenced me or affected me in a certain way. Perhaps it may be something in the subconscious, but I am unaware"*

6-The residents were not able to differentiate between Balint sessions and support groups. They expected the Balint session to be focused on their psychological well-being.

*"I feel that the patient is always the first during the encounter; at least in this space, we need to be the first; I feel sometimes that is lost."*

*"Balint needs to adapt to what we are going through; the goal of the Balint is to help our psychological well-being."*

*"I know it has to do something with empathy and putting yourself in the shoes of the patient, but sometimes we want our peers or supervisors to put themselves in our shoes first; it should be about us first; I sometimes feel that lost; the focus should be us."*

*"I want this space to be selfish; it is ok to be selfish sometimes."*

After the modified BG, the focus group revealed that the residents were aware of the change during the last 10 min and considered it a closure. The closure helped in decreasing some of the uncertainty in Balint seminars. Most of the residents did not consider the change helpful.

*"we used when we finish, we have lots of questions at the end as if we have not done anything; maybe this is better in the new format."*

*"I noticed those who presented have felt a closure to the case."*

*"the closure is the only new change. The moderators are participating in the Balint session at the end of the session, and it is good to have a closure to the case."*

*"there was particular uncertainty left at the end of the session; it is decreased; since modified, you don't get the correct answer; this is not the goal anyway; there is still uncertainty, but it is much less than before."*

Few residents considered that change helped make them more aware of their emotions and feelings but did not help in their perception that they have achieved the goals of Balint. The input of the facilitators was helpful regarding the description of the thought process and its relation to emotions.

*"We are more aware of our emotions; this has improved."*

*"We knew the goal of the Balint session was to improve the patient and sense the patient more and improve the doctor-patient relationship and my feelings towards the patient. Did the change or Balint help me achieve this? I don't think so at this level."*

*"I think describing the thought process at the end of the 10 minutes is what we should be able to reach at the end of Balint after multiple sessions. I think this is what we tried to do and achieve to a certain degree in the last 10 minutes: how to think and feel. We can still work on this, but it was a good start."*

They again voiced the need for support groups more than Balint seminars. *"Balint is hot chocolate on a hot summer day, the right drink but not the right time."* They asked that the session be optional and less rigid. Moreover, they preferred direct feedback on their performance during a real patient encounter over Balint seminars. They felt the sessions were artificial as they struggled to select a case. It is unlikely to find a patient "stuck" in their mind, mainly because the case would have already been discussed with a preceptor or peer or resolved over time.

*"I think for it to be a teaching moment, it has to happen at the spot when it happened."*

### Empathy and psychological skills

A total of 14 residents completed all the surveys. A two-way ANOVA was conducted to examine the effect of the Balint training type on the Jefferson Physician Empathy scale across time. There was no statistically significant interaction between the effects of time and training method in the empathy scores of family medicine residents, $F(1,13) = 0.302$, $p = 0.60$. Similarly, there was no statistically significant main effect of time on empathy

scores, F(1,13) = 2.312, $p$ = 0.15. There were no differences between empathy scores and training method, F(1,13) = 0.026, $p$ = 0.87.

A two-way ANOVA was conducted to examine the effect of the Balint training type on the PMI score across time. There was no statistically significant interaction between the effects of time and training method in the PMI scores of family medicine residents, F(1,13) = 0.009, $p$ = 0.09. Similarly, there was no statistically significant main effect of training on PMI scores, F(1,13) = 1.039, $p$ = 0.07. However, there was a statistically significant main effect of time on PMI scores, F (1,13) = 7.709, $p$ = 0.02.

## Patient perspective

A total of eight residents agreed to approach their patients. A total of 72 and 73 patients were included during the traditional BG and the modified BG, respectively. Patients were predominantly females (91/144, 63.2%) and equally distributed between married (59/144, 40.7%) and single (53/144, 53.1%). The distribution of the patients was similar across the traditional and modified BG. There was a variety of acute complaints and the presence of chronic diseases. The mean CARE score was 45.8 (SD = 5.5) and 45.5 (SD = 6.3) during the traditional and modified BG, respectively. A two-way ANCOVA was conducted to examine the effect of Balint training type and resident on the CARE scale. There was no statistically significant interaction between the effects of resident and training method in the CARE scores, F (7,129) = 0.689, $p$ = 0.68. Similarly, there was no statistically significant main effect of the training method or resident on PMI scores.

## DISCUSSION

A modified BG was developed to respond to the needs and concerns of residents regarding Balint groups' uncertainty. The modified BG's effectiveness was evaluated by measuring the residents' empathy through self and patient evaluation, residents' psychological skills, and satisfaction with the educational activity through focus groups, as guided by the Kirkpatrick broad classes of program outcomes. This study has shown that the residents viewed the modified BG as a closure at the end of the session. It helped decrease some of the uncertainty in Balint seminars and guided the resident regarding describing the thought process and its relation to emotions. Few residents considered that the change helped them be more aware of their emotions and feelings but did not help in their perception that they have achieved the goals of Balint.

The modified BG did not improve the residents' empathy or psychological skills. Although the residents were aware of the goals of BG, they considered them theoretical and could not see the application in actual practice. They preferred immediate feedback on encounters with difficult patients rather than discussing them in Balint seminars. Other residents considered the benefits long-term, subtle and subconscious. Accordingly, the benefit may not appear using qualitative methods and validated instruments. This common theme among Balint research is that qualitative research suggests improved personal and professional development, whereas quantitative findings point to a potential nonconclusive value of BS (*Hamadeh, Antoun & Romani, 2020*; *Salter et al., 2020*; *Van Roy, Vanheule & Inslegers, 2015*; *Yazdankhahfard, Haghani & Omid, 2019*).

## Strengths and limitations

Due to the lack of extensive research on modified Balint sessions (*Van Roy, Vanheule & Inslegers, 2015*), this study is unique in evaluating the impact of BG on psychological skills and empathic abilities from the patient's perspective in addition to self-reported measures. Another strength is the triangulation of the data points from focus groups, various self-reported measures, and patient-reported measures. Nevertheless, the small sample size is a significant limitation that could have resulted in non-significant results. Balint seminars are not commonly utilized in Lebanon as compared to the US. There are no other institutions that delivered Balint to increase the sample size. Our department's cohort is small each year, so all residents attend the same BG. This could lead to a bias in their maturity and level of skills. Nevertheless, the pre-post design and the accounting of time in the analysis should alleviate this bias.

Moreover, cultural issues may play a role, so we cannot extend the sample to residents in the US. Furthermore, the research was conducted during the COVID pandemic, and the country passed through a financial crisis. The residents may have suffered from burnout and psychological distress that led them to voice their needs for support groups rather than Balint groups, where the focus is on the residents' well-being rather than the patient. The discussions during the Balint groups may have been channeled to address the current struggles in the country, and residents brought themselves as victims/patients rather than seeing themselves as a healer. A previous study has shown that the social/political/religious context distracts doctors from fulfilling their professional role (*Antoun et al., 2019*).

## Future implications

This study is an eye-opener about the need to re-evaluate the current Balint process that Michael Balint initiated in 1955. A simple leader debriefing at the end of the session has improved the perception of the residents about Balint's uncertainty and ambiguity. Other modifications may include reminding the residents of the purpose of the Balint seminars now and then and not just at the beginning of the year. A more extended intervention beyond 6 months may be needed to see the effect of the new intervention. Further research should attempt to compare Balint seminars with other methods of making meaningful connections with patients, such as narrative medicine workshops (*Lijoi, 2023*).

## CONCLUSIONS

The facilitators' debriefing at the end of BG about key themes may help give the residents closure of the medical encounter, decrease the uncertainty, and make them more aware of their feelings. Nevertheless, Balint groups are still not very well accepted by the residents, and they prefer direct feedback and support groups. Using validated instruments, BG was ineffective at improving the residents' empathy and psychological skills. Further research should aim at other modifications of the Balint process and compare it with other venues that allow the residents to talk about patients, such as support groups or narrative medicine.

## ACKNOWLEDGEMENTS

The authors acknowledge Dr. Natally AlArab for her contribution to the data collection.

### Funding

This work was supported by the Centre of Teaching and Learning, American University of Beirut, award number (103965). The funders had no role in study design, data collection and analysis, decision to publish, or preparation of the manuscript.

### Grant Disclosures

The following grant information was disclosed by the authors:
Centre of Teaching and Learning, American University of Beirut: 103965.

### Competing Interests

Jumana Antoun is an Academic Editor for PeerJ.

### Author Contributions

- Jumana Antoun conceived and designed the experiments, analyzed the data, authored or reviewed drafts of the article, and approved the final draft.
- Beatrice Khater conceived and designed the experiments, performed the experiments, analyzed the data, authored or reviewed drafts of the article, and approved the final draft.
- Hala Itani conceived and designed the experiments, performed the experiments, authored or reviewed drafts of the article, and approved the final draft.
- Jihane Naous conceived and designed the experiments, authored or reviewed drafts of the article, and approved the final draft.
- Maya Romani conceived and designed the experiments, authored or reviewed drafts of the article, and approved the final draft.

### Human Ethics

The following information was supplied relating to ethical approvals (*i.e.*, approving body and any reference numbers):

The American University of Beirut granted ethical approval to carry out the study within its facilities (Ethical application ref:#SB-201900379).

### Data Availability

The raw data are available in the Supplemental Files.

### Supplemental Information

Supplemental information for this article can be found online at http://dx.doi.org/10.7717/peerj.15279#supplemental-information.

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
