# Peer review of "Effectiveness of a modified Balint group process on empathy and psychological skills employing Kirkpatrick's evaluation framework"

_PeerJ, doi:10.7717/peerj.15279_

## Round 0.1 · original submission · Major Revisions

Dear authors,

Please go through the comments from Reviewers and address these. Please submit the revised version at the earliest preferably within 10 days.

regards,
Animesh

Reviewer 1 ·

Basic reporting

The manuscript is reasonably well written, but the Kirkpatrick framework is barely discussed. Why this evaluation model over other possible approaches?

Experimental design

It would be useful to list hypotheses and link analyses to testing respective hypotheses. Talk about the possible limitations of such a small sample size.

Validity of the findings

No comment

Additional comments

It would strengthen the manuscript if additional research recommendations could be offered and some idea of what the theoretical and empirical contributions might be.

·

Basic reporting

- Clear, unambiguous, professional English language used throughout.

It’s all right for me but is the word « closure » the right word ? and if so, is the debriefing always a closure ?

- Intro & background to show context.

It’s ok but (small detail) in the background (and even more in the abstract) I’m not sure it’s ok to write in the first line « BG » and not « Balint group ».
Idem for PMI when first stated.

- Literature well referenced & relevant.

For me :
« Buffel du Vaure C, Lemogne C, Bunge L, Catu-Pinault A, Hoertel N, Ghasarossian C, Vincens ME, Galam E, and Jaury P. 2017 « Promoting empathy among medical students : A two-site randomized controlled study » J Psychosom Res 103 : 102-107. »
is more relevant and appropriate than the preliminary research :
« Airagnes G, Consoli SM, De Morlhon O, Galliot AM, Lemogne C, and Jaury P. 2014. Appropriate training based on Balint groups can improve the empathic abilities of medical students : a preliminary study. J Psychosom Res 76 :426-429. 10.1016/j.jpsychores.2014.03.005 »

- Raw data supplied.

I didn’t find the raw data of the CARE scale and can you give the questions of Q1 to Q20 and how you code them ?

Experimental design

- Research question well defined, relevant & meaningful.

Yes, very interesting and it is important to have patients in the evaluation and to have quantitative and qualitative research at the same time to be more complete.

- It is stated how the research fills an identified knowledge gap.

yes

- Rigorous investigation performed to a high technical & ethical standard.

For the quantitative part, it needs at least a control group and of course more students

- Methods described with sufficient detail & information to replicate.

Not completely :
-« All post-graduate years 2, 3, and 4 attend the sessions » obligatory ? not the same level of study ?
-how many BG during the first 6 months ? and the next 6 months ?
-When do you explain what you expect from these BG ? by oral at the beginning of the first BG ? Is it written somewhere ?
-is there, during the same time of the study an other place where these student could have theory about relation and or communication ? and another place where they can talk about their patients ?
-when was the CARE scale done ?
- It would be interesting and more comprehensive for readers to have a figure showing different times of the study and when the different scales and focus group are done.

Validity of the findings

-How can you explain the results of resident 5, 6 and 7 for the JFSE ? Have they patients in the study ?
So, can you make a sensitivity analysis without these 3 results ?

-There must be a mistake in these two paragraphs :
Empathy and Psychological Skills
A total of 14 residents completed all the surveys. A two-way ANOVA was conducted to examine the effect of the Balint training type on the Jefferson Physician Empathy scale across time. There was no statistically significant interaction between the effects of time and training method in the empathy scores of family medicine residents, F(1,13) = 0.302, p = 0.60. Similarly, there was no statistically significant main effect of time on empathy scores, F(1,13) = 2.312, p = 0.15. There were no differences between empathy scores and training method, F(1,13) = 0.026, p = 0.87. 266 267
A two-way ANOVA was conducted to examine the effect of the Balint training type on the Jefferson Physician Empathy scale across time. There was no statistically significant interaction between the effects of time and training method in the PMI scores of family medicine residents, F(1,13) = 0.009, p = 0.09. Similarly, there was no statistically significant main effect of training on PMI scores, F(1,13) = 1.039, p = 0.07. However, there was a statistically significant main effect of time on PMI scores, F (1,13) = 7.709, p = 0.02.

- For this kind of study I’m not sure that you can compare students who are not in the same post graduate year.

- Conclusions

« this study is unique in evaluating the impact of BG on psychological skills and empathic abilities from the patient perspective in addition to self-reported measures »
This is not completely true = read : « Buffel du Vaure C, Lemogne C, Bunge L, Catu-Pinault A, Hoertel N, Ghasarossian C, Vincens ME, Galam E, and Jaury P. 2017 « Promoting empathy among medical students : A two-site randomized controlled study «

Additional comments

The design is clever and interesting ; infortunately the weak number of participants makes the quantitative results difficult to interpret

---

## Round 0.2 · accepted · Accept

Congratulations. Editorial office team will get in touch with you for further processing and publication formalities.